# One-session treatment for specific phobias: Barriers, facilitators and acceptability as perceived by children & young people, parents, and clinicians

**Emily Hayward**[1☯¤a*], **Kiera Solaiman**[2☯¤b], **Penny Bee**[3☯], **Amy Barr**[2☯¤c], **Hannah Edwards**[1☯¤d], **Jennifer Lomas**[2☯¤e], **Lucy Tindall**[1¤a‡], **Alexander J. Scott**[2¤f‡], **Katie Biggs**[2‡], **Barry Wright**[1,4‡]

**1** Child Oriented Mental health Intervention Centre (COMIC), Leeds & York Partnership NHS Foundation Trust, York, United Kingdom, **2** School of Health and Related Research (ScHARR), University of Sheffield, Sheffield, United Kingdom, **3** School of Nursing, Midwifery & Social Work, University of Manchester, Manchester, United Kingdom, **4** Hull York Medical School, University of York, York, United Kingdom

☯ These authors contributed equally to this work.
¤a Current address: Department of Health Sciences, University of York, York, United Kingdom
¤b Current address: Department of Psychology, University of Sheffield, Sheffield, United Kingdom
¤c Current address: National Institute for Health and Care Excellence, Medical Technologies Evaluation Programme, Manchester, United Kingdom
¤d Current address: Psychological Services, Cambridge and Peterborough NHS Foundation Trust, Cambridge, United Kingdom
¤e Current address: Kirklees Council, Kirklees, United Kingdom
¤f Current address: Department of Psychology, Keele University, Keele, United Kingdom
‡ These authors also contributed equally to this work
* e.hayward@york.ac.uk

**Data Availability Statement:** While the data presented does not contain any direct identifiers, due to the sensitive nature of the data presented

## Abstract

Between 2015 and 2020 the Alleviating Specific Phobias Experienced by Children Trial (ASPECT) was conducted in the UK to examine the non-inferiority of One-Session Treatment in comparison to Cognitive Behavioural Therapy based interventions for children and young people with specific phobias. A nested qualitative evaluation was conducted as part of this trial to examine the acceptability of One-Session Treatment. Qualitative interviews were conducted with children and young people taking part in the trial, their parents/guardians, and clinicians delivering the intervention, about their experiences and the acceptability of One Session Treatment. Interviews were digitally recorded and transcribed verbatim. Analysis followed a qualitative framework approach, a widely used method of analysing primary qualitative data pertaining to healthcare practices with policy relevance. Stakeholder groups found One Session Treatment to be an acceptable intervention and barriers and facilitators for its implementation into services were also identified. Potential barriers included challenges to patient flow and treatment scepticism, whilst facilitators included adopting a child-centred approach, child readiness and suitability, opportunity for increased momentum, parental support and involvement, and proximal and distal gains. For One Session Treatment's successful implementation into services, consideration of these barriers is needed and suitability guidance for its use in this population should be developed. Further research exploring children and young people's experiences of receiving Cognitive

and vulnerable participant group, we would not be able to make the dataset publicly available due to ethical reasons. Any requests for sharing of data can be made to the corresponding author or Sheffield Clinical Trials Unit (ctru@sheffield.ac.uk). The ASPECT trial management team will consider the sharing of data on a case-by case basis in line with the ethical approval and patient information sheets.

**Funding:** This research was funded by the National Institute for Health Research under its Health Technology Assessment Programme (Grant Reference Number: HTA 15/38/04). The views expressed are those of the author(s) and not necessarily those of the NIHR or the Department of Health and Social Care. The funders had no role in study design, data collection and analysis, decision to publish, or preparation of the manuscript.

**Competing interests:** The authors declare that there is no conflict of interest.

Behavioural Therapy and its acceptability in comparison to One Session Treatment would be welcomed.

## Introduction

Specific phobias are characterised as an intense fear and avoidance of a specific object or situation [1]. Prevalence estimates suggest 5–10% of children and young people (CYP) experience specific phobias [2]. Phobias impact on many day-to-day activities [3] and can continue to impact in the long term if left untreated [4]. Despite this, fewer than 10% of CYP seek treatment [4]. Cognitive Behavioural Therapy (CBT) is a common therapeutic modality to treat specific phobias and is supported by extensive evidence of its efficacy [5–7]. Delivered over 6–12 sessions, CBT combines several techniques including graduated exposure to phobic stimuli, cognitive reframing and restructuring, and relaxation techniques.

There have been significant changes to the provision of Child and Adolescent Mental Health Services (CAMHS) in the United Kingdom following the government's Green [8] and Future In Mind [9] papers, occurring at a time when waiting lists to access services are long and availability of CBT and phobia treatments are limited. CBT is time intensive [10,11] and expensive [12,13] which has promoted further research into other briefer yet accessible treatment models and options [14,15], including those for specific phobias in primary care. One such option is One-Session Treatment (OST) [16]. Employing many of the same techniques as CBT, OST is delivered over two appointments: one assessment session and one extended graded exposure session lasting up to 3 hours. The Alleviating Specific Phobias Experienced by Children Trial (ASPECT) aimed to assess the clinical and cost-effectiveness of OST in comparison to CBT [17]. ASPECT found OST to be non-inferior and potentially superior to traditional CBT, as well as cost-effective [18,19].

The development of novel psychological interventions demands treatments that are both effective and suitable for stakeholder's needs [20]. This paper adopts a qualitative methodology to report on the acceptability of OST from multiple stakeholder perspectives, specifically CYP, their parents/guardians, and clinicians who deliver the treatment. This is important in furthering our understanding of OST and its implementation in the real world. We aim to bring together the experiences of these groups, distilling their narratives into key recommendations for optimising and implementing OST in routine practice.

## Materials and methods

### Research design and setting

This was a qualitative descriptive study exploring the experiences of those who were involved in OST sessions as part of ASPECT. ASPECT had 13 sites across England, comprising 12 NHS trusts (including 26 CAMHS sites and 3 affiliated voluntary agency services) and one university-based CYP wellbeing service.

### Sampling

The eligibility criteria for ASPECT was CYP aged 7–16 years, who were experiencing a specific phobia defined by DSM-V criteria assessed using the Anxiety Disorder Interview Schedule (ADIS). CYP were excluded or withdrawn from the trial if exposure therapy was not the best available or first line treatment or had potential to be unsafe or cause harm.

We interviewed three separate groups; 1) CYP who were randomised to receive OST; 2) their parents/guardians; and 3) clinicians who had delivered OST. Maximum variation sampling was used to ensure a diverse sample, including participants based in different locations and with different ages, gender, and phobia types (See Tables 1–3). 35 parent-child dyads were approached with 27 agreeing to participate. Those who refused cited a lack of time or the researcher was unable to contact them. All clinicians approached took part in an interview. The eventual sample size was achieved once data saturation had been reached and no new themes, ideas, or concepts were surfacing. Data saturation was determined by team discussion and an additional 3 interviews with CYP and parents/guardians undertaken after this point to confirm saturation had been reached.

## Intervention

OST is a variant of CBT, using many of the same techniques (e.g. skills training, cognitive challenges, participant modelling, reinforcement, and graduated exposure), however, it takes a more condensed approach. Unlike CBT, which is delivered over weekly sessions with accompanying homework in between, those randomised to OST attended an initial functional assessment session (lasting up to 1 hour) used to plan treatment followed by an exposure session (lasting approximately 3 hours). OST delivery followed a treatment protocol used in previous research [21].

## Data collection

Interviews were conducted with CYP and their parents after they had completed the final outcome measure of the trial, six months after randomisation. Clinician interviews were conducted when their involvement in the trial was finished. Interviews were conducted by six trained research assistants working on the ASPECT trial, including authors EH (BSc), AB (MSc), JL (MSc) and HE (MSc). All interviewers had experience working with CYP, parents/guardians and clinicians and were trained in administering and analysing qualitative

**Table 1. CYP demographic information.**

| CYP Demographic Information Total (n = 27) | |
|---|---|
| **Age** | **Total** |
| 7–8 years | 5 |
| 9-10years | 8 |
| 11-12years | 8 |
| 13–14 years | 0 |
| 15–16 years | 6 |
| **Sex** | **Total** |
| Male | 9 |
| Female | 18 |
| **Phobia Type** | **Total** |
| Animal type | 12 |
| Natural Environment type | 1 |
| Blood-Injection Injury type | 8 |
| Situational type | 0 |
| Other type | 6 |
| **Ethnicity** | **Total** |
| White (English / Welsh / Scottish / Northern Irish / British) | 27 |

**Table 2. Parent demographic information.**

| Parent Demographic Information Total (n = 27) | |
|---|---|
| **Age** | **Total** |
| 30–39 years | 8 |
| 40–49 years | 12 |
| 50–59 years | 7 |
| **Sex** | **Total** |
| Male | 0 |
| Female | 27 |
| **Ethnicity** | **Total** |
| White (English / Welsh / Scottish / Northern Irish / British) | 27 |

interviews prior to commencing data collection. Five interviewers were female, and one was male. Some participants were familiar with their interviewer and had previously met at baseline and follow-up data collection visits. Depending upon participant preference, interviews with parents and older CYP (e.g., 12–16 years) were either conducted face-to-face, or via the telephone. All interviews with younger CYP (e.g., 7–11 years) were face-to-face. Parent-child dyads were interviewed separately, apart from cases where younger CYP requested for a parent to be present. In these cases, parents were instructed by the researcher not to speak while the interview was in progress. Interviews were conducted at a mutually convenient location (i.e. participant homes or treatment settings). At the time of the clinician interviews, COVID-19

**Table 3. Clinician demographic information.**

| Clinician Demographic Information Total (n = 16) | |
|---|---|
| **Role** | **Total** |
| CYP PWP | 4 |
| CAMHS Practitioner | 2 |
| CBT Therapist | 2 |
| Clinical Psychologist | 2 |
| School Wellbeing Worker | 2 |
| Consultant Clinical Psychologist | 1 |
| Junior Doctor | 1 |
| Psychiatrist | 1 |
| Senior Youth Counsellor | 1 |
| **Gender** | **Total** |
| Female | 81% |
| Male | 29% |
| **Years of experience delivering psychological interventions to CYP** | **Total** |
| 5 years or less | 53% |
| 5–10 years | 27% |
| 10–20 years | 7% |
| Over 20 years | 13% |
| **Treatment preference before ASPECT** | |
| OST | 60% |
| CBT | 13% |
| No preference | 27% |

pandemic restrictions were in place and interviews were conducted over the phone as opposed to in clinical settings. In collaboration with Patient and Public Involvement (PPI) representatives, the research team developed topic guides to explore the feasibility and acceptability of OST. These topic guides enabled interviews to follow a semi-structured format (See S1–S3 Files).

Interviews with parents and CYP focussed on their experiences of the phobia; the impact on them personally and wider family impact; their perceived treatment need; expectations of treatment and their engagement in treatment and acceptability. Questioning was adapted in line with CYP's developmental age and understanding, with younger children's interviews adapted to include the 'draw and write' technique [22,23] to scaffold discussion around their experiences where needed. CYP interviews lasted up to 30 minutes, and parent interviews lasted up to 60 minutes.

Clinician interviews followed a similar format, lasting up to 60 minutes. These interviews focussed upon professionals' experiences of delivering OST; potential barriers and enablers to its implementation into practice; the support required by individuals; team and organisation and their perceptions around OST's suitability for CYP with phobias.

### Data analysis

Encrypted digital recorders were used to record interviews, and these were then transcribed verbatim following participant consent. No notes were taken during the interviews. A qualitative framework approach [24] was employed and coding was completed independently by six trained researchers. Researchers met regularly to develop a shared coding framework as new themes emerged. Transcripts were coded using NVivo Version 12 [25]. A spreadsheet was created to consolidate emerging framework themes, which was regularly reviewed and developed to incorporate new codes and remove codes that were no longer relevant. The spreadsheet also included demographic information to keep track of any emerging patterns specific to demographic groups and to monitor sample diversity. Following this, a finalised framework representative of the data set was produced. Findings were presented to PPI representatives once during the analysis phase to confirm validity, coherence and conceptual relevance.

### Ethics

Health Research Authority and Research Ethics Committee approval was gained in February 2017 from North East–York Ethics Research Committee (ref: 17/NE/0012). All participants provided written informed consent to take part in the audio-recorded interviews.

### Results

Results from the main trial suggested that OST was non-inferior to CBT and was clinically and cost-effective [18,19]. There were no obvious differences apparent in demographic characteristics. Good levels of acceptability were found across the three samples and several facilitating factors were identified: Child-centred approach, child readiness & suitability, opportunity for increased momentum, parental support & involvement, proximal & distal gains. Despite this, distinct barriers were noted: challenges to patient flow and treatment scepticism. Narratives around these will be discussed in turn, primarily focusing on shared themes across the participant groups. While some variation in themes across participant groups were present (see S4 File), this article will specifically focus on commonalities prominent in provision and implementation of OST into services.

## Perceived barriers to OST implementation

**Challenges to patient flow.** A consistent theme that was present in narratives across all three samples was that access to treatment was difficult due to minimisation of the phobia's impact. Even when CYP's functioning was impaired, barriers were still encountered in accessing treatment. Parents also reflected on their own role in this in terms of overlooking their child's phobia initially, believing it was just a phase their child was experiencing.

*"They would just tell me that it's just an insect you don't need to worry about it."*

[Male, 9 years old, Bees/Insect Phobia]

*"I always thought he'll grow out of it, he'll grow out of it but like you say- well then at 10 he still hadn't grown out of his phobia."*

[Parent/Guardian, 12 year old girl, Insect Phobia]

When reflecting on why they had not sought help sooner, there were assumptions among parents that their child would be rejected due to health services' low prioritisation of phobias and perceptions of a lack of funding in children's mental health care. Some parents continued to manage the phobia themselves while others were just not aware of the existence of support for this. This was also reflected in CYP's narratives where there also appeared to be a low awareness of phobia-specific treatment options that may be available to them.

*"I mean it didn't even occur to me to go to a doctor and ask for help. It hadn't occurred to me at all, I think we were trying to manage it ourselves and I think it's why it's that, having that frustration and then realising actually its, its beyond what we thought"*

[Parent/Guardian, 12 year old girl, Dog Phobia]

*"No, 'cause I didn't know it was available, so I didn't even know it was a thing. So, I'd never gone to anyone. Even people I'd spoken to about it, I hadn't even realised it was a thing."*

[Female, 16 years old, Vomit Phobia]

When discussing experiences of attempts to seek treatment for their child's phobia, parents explained how they were often rejected by services. Clinician's narratives also highlighted how this was not an uncommon experience and very often phobias would not meet the referral criteria for services. They also confirmed that phobias may not be referred as such and were unclear on where these CYP would be seen. Some clinicians even noted that prior to their involvement in the trial, they were unaware that CAMHS did accept and treat phobias.

*"Yeah I did, I think it's a shame how it works in the [Locality Name] in terms of erm- a lot of these phobia referrals wouldn't get accepted but then it isn't quite clear where they do go."*

[Clinical Psychologist, CAMHS]

*"Everywhere we turned they just said they weren't interested basically and then eventually we you know- we got given the details of yourselves and the trial that you were doing."*

*[Parent/Guardian, 7 year old girl, Dog Phobia]*

**Treatment scepticism.** Another barrier identified across the sample was a general lack of understanding of OST and its underlying mechanisms. Gaps in clinician understanding were

reported and they commented on feeling unsure about how the intervention worked and had concerns around whether it would be effective for CYP. Perceptions from CYP in terms of their understanding and engagement was motivated by a lack of opportunity costs rather than an expectation that OST would be successful.

*"I was a little bit sceptical, maybe? [Laughs] Erm, I wasn't really sure how it would go and how, erm- how engaged, err, the young people would be in the process, really."*

[Psychiatrist, CAMHS]

*"I dunno, I just wanted to know how on earth that was- what's going to happen to make me not scared of them as much. I didn't think it would make a difference but yeah."*

[Female, 10 years old, Needle Phobia]

Narratives from parents suggested they took a cautious approach to OST as a treatment. This seemed to be motivated by a desire to protect their child and themselves from disappointment. As such, they framed their expectations in the context of improvement as opposed to a complete cure.

*"I was quite realistic with it because I thought "You can't cure it in three hours". I understand the idea around it, but I was just hoping it would give him more confidence in how to deal with it. So, I was hoping that he would feel stronger. . . So, that's what I wanted out of it, was coping mechanisms, but I didn't have any- I didn't want to set myself up and think "He's gonna be absolutely fine when he walks out" because that's just upsetting for both of us. So I thought I'll be practical and I'll hope it gives him the tools he needs, but I didn't think it would cure him."*

[Parent/Guardian, 10 year old boy, Blood Phobia]

### Perceived facilitators for OST implementation

**Child-centred approach.** An important and valued aspect of OST from the perspectives of all three groups was around ensuring CYP played an active role in their therapy experience. CYP valued and appreciated feeling they were in control while developing their fear hierarchy and during their exposure session. They also noted the importance of the way clinicians communicated with them at an appropriate level to their developmental stage. This was also reflected in parent narratives around their child's experience of OST as they appreciated clinicians' efforts to create an equal dynamic when working with their child. Clinicians also discussed this in their interviews in the context of working collaboratively with CYP and allowing them to take the lead in their progression. This supported the development of a positive therapeutic rapport.

*"I think it wasn't like- I was you know it wasn't like I was being treated like a child. And it was- so she somehow managed to not make me feel like I was being treated as a child but she also managed to support me in the way that I needed to be supported if that makes sense?!"*

[Female, 15 years old, Vomit Phobia]

*"He's just- he just- patience that was the thing he had patience and he got down to [Child Name]'s level a lot instead of like towering over him talking down. But he actually got down and was speaking to him and that was the thing he didn't tower over him"*

[Parent/Guardian, 7 year old boy, Dog Phobia]

*"So in actual fact there's quite a lot of hours that you end up spending with the family. So- so you sort of build that trusting relationship and safe space. It's really you know- you make sure it's really collaborative and check out everything. So I think sort of in that sense you know it- it didn't feel like I was- I suppose the risk is it feels like your sort of doing, you know doing an intervention with somebody before you've developed that trusting relationship, but it didn't really feel like that. I felt like we had enough hours together to kind of build that really"*

[Clinical Psychologist, CAMHS]

**Child readiness & suitability.**   Across all three samples, there was a strong message around CYP's readiness to access treatment in the context of developmental stage and intrinsic motivation. Narratives from CYP centred on their ability to understand the impact of their phobia on their life and being in a place emotionally to address this. This was also mirrored in parent narratives; however, parent's main concerns were around their child being resilient enough to engage in the intensive nature of the treatment. Clinicians discussed both aspects in their interviews and picked up on other elements of suitability such as presence of any comorbidities that may impact the child's ability to engage in the treatment.

*"And has to realise that it is a lot–it has to be at the right time I think, as well. It's hard when you've got a lot going on in your life–it's hard to focus your time and energy into overcoming the phobia because it does take a lot out of you. Um. 'Cause all the energy you'd usually be using to worry about it you're kind of having to force yourself to go "no, you're okay–you don't have to worry" you're having to kind of like actively think about it. So, it does take a lot, so I think it has to be at the right time with the right person, cause you just kind of go, gone into the session, you're sort of bombarded with loads of information, which isn't necessarily a bad thing, and then you have to take that and apply it to your life (yeah)"*

[Female, 16 years old, Vomit Phobia]

*"It's hard work and it's intense and it's draining and you're tired after it. I was tired even though I wasn't doing it. So I think it's a lot for- for the child to do. Um, and I think the child has to really, really want to sort the phobia out or they won't commit to it, cause it's quite a lot of work in three hours"*

[Parent/Guardian, 10 year old boy, Blood Phobia]

*"But beginning, it's the beginning bit when you're giving more about the psychoeducation and you're sort of trying to do a bit of the detective work with them. For those children who maybe have got tics but then also a history of trauma or some learning difficulties they actually need a little bit longer to actually understand the treatment itself and engage with you as a therapist and trust you. So that would be my sense with OST is that you may just need- you might not be able to do your assessment in one session, you might need to do that over two or three sessions but you then might do the treatment bits still in that extended treatment session."*

[Consultant Clinical Psychologist / Core CAMHS Clinical Lead, CAMHS]

**Opportunity for increased momentum.**   Narratives from CYP and clinicians when reflecting on the acceptability of OST focused on the momentum the extended exposure session allowed them to gather. There was a sense that this allowed them to capitalise on earlier progress and this motivated them to get further on their anticipated goals on their hierarchy. CYP and clinicians also discussed their perceptions of reduced barriers to accessing OST (e.g.

less clinic visits, reduced build-up of anxiety, and reduced opportunity for avoidance) which helped to increase their motivation in engaging with the treatment. Parents and guardians also acknowledged this, particularly in the context of reduced logistical barriers (e.g. taking time out of work, reduced time off of school to bring children to appointments) but also acknowledged the cost of the intensive nature of the session, which would be present in CBT but more dispersed over the number of sessions.

*"Well, I dunno, cause I feel like CBT, obviously I haven't went through it so I don't fully know, but because it's like one hour sessions per, I don't know if like, in-between the weeks you'd sort of feel like you'd lost some of the progress and you'd just be starting from square one each week. . . . . ..And I thought that like with the OST, it's three hours, you get it done and the stuff you get done is more likely to be solidified, so if it falls back to square one, you're at least somewhere further in the therapy."*

[Male, 15 years old, Needle Phobia]

*"When I was running the OST, say for example, there was a spider phobia one that I did and we were sort of like two hours into it. I was thinking god you know, ordinarily this would have been, you know three or four sessions to get to this point and here we are two hours down the line going big leaps. Sort of thinking about benefits in terms of you know cutting down on waiting times etcetera and also cutting down on the anxiety that that person maybe building with regards to receiving treatments in the times that they're waiting."*

[Primary Care Mental Health Worker, CAMHS]

*"I think from an expectation point of view I was- I was pleased she was put into that arm because I was very conscious that if it's a weekly thing then it was going to disrupt her schooling, quite significantly. It was going to disrupt her schooling because although it was probably only going to be an hour session, you've got to travel to and from, it's a lot of time out of school in your GCSE years so I was pleased from that point of view that was obviously not going to happen."*

[Parent/Guardian, 15 year old girl, Needle Phobia]

**Parental support & involvement.** For this theme narratives around parental presence were varied. Some had a preference for parental presence, others did not. This was determined on a case-by-case basis for CYP and parents and a pattern did not appear to be present. Narratives around parental presence during therapy sessions followed the theme of having choice. CYP felt that this should be an option that was given but not a mandatory component. This was also reflected in parent narratives and was supplemented by the need to have a good handover to ensure they could continue to complete exposure work outside of therapy to maintain therapeutic gains. Clinicians picked up on both themes in their narratives and highlighted that this would need to be decided on a case-by-case basis and would advise against parental presence if the parent was identified as a safety behaviour or maintaining factor in their child's phobia.

*"I mean for yeah for me it was useful kind of having someone there 'cause of course I don't know [Therapist Name] that well I don't- didn't know the nurse that well. So having someone that I know really well as as support is something really useful for me anyway."*

[Female, 15 years old, Needle Phobia]

*"Yeah, I think it's good psychology and I think it's good for parents to understand why a child's maybe behaving in the way that they're behaving rather than just going "Oh, you're just mis-behaving". Actually, what's the cause behind that? So, yeah, I thought it was really good that I could actually get involved. Even though you weren't doing anything and you didn't say any-thing 'cause you didn't want to interfere, just the fact that you could listen to it and hear it."*

[Parent/Guardian, 10 year old girl, Needle Phobia]

*"I probably would have made the decision as a service that in having the parent in the room we were inadvertently maintaining the anxiety if the parent felt like a reassuring presence."*

[Psychological Wellbeing Practitioner, University-based IAPT Research Clinic]

**Proximal and distal gains.**   Narratives from children and parents suggested they had not only improved self-management in the context of the phobia but had also gained a lot of knowledge. This was not something they foresaw at the outset of starting treatment but felt it was something they found helpful on reflection. This was also reflected in clinician views but focused more on secondary distal gains in the context of CYP's functioning and more general-ised anxiety management strategies.

*"I still get quite worked up about come quite distressed about my mocks because of my GCSEs and they're stressful but I was doing some breathing this morning as well kind of just going up on my way to school cos getting quite worked up and it is it was the kind of the same thing where I went, I was quite nerv- no before my exam, I went through my exam it wasn't as bad as I thought it would be and then I've calmed myself down at that point so it yeah, useful things"*

[Female, 15 years old, Needle Phobia]

*"An understanding of what's actually going to happen. Not just a needle goes into your arm, but actually what your brain's telling you. What's that doing to your body and why do you feel sick and why do you start feeling dizzy? I think all of those things kind of helped her under-stand why she goes through the feelings that she goes through and what she can do to stop them and the fact that she can say "Yes", "No", "Can you change this? This'll make it easier for me".*

[Parent/Guardian, 10 year old girl, Needle Phobia]

*"Erm, I think, err- and certainly one of the cases, erm, was open to CAMHS, erm, kind of for ongoing work around a different anxiety disorder, I think alongside the phobia. And just- just generally seemed to have a- a reduction in her anxiety levels, even though it kind of, you know, once this- the phobia side of things had improved. So, it seemed to have kind of wider effect on her, her anxiety levels."*

[Psychiatrist, CAMHS]

## Discussion

The aim of this study was to explore the acceptability of OST from the perspectives of the CYP who received it, their parents/guardians, and clinicians who delivered it. Our findings suggest that all three stakeholder groups found OST to be an acceptable treatment, and CYP experi-enced proximal and distal gains in the context of managing their phobia and other areas of

functioning. OST supported parents and CYP in developing a better understanding of the mechanisms behind the phobia and transferable skills in managing anxiety more generally. There was a narrative from all three stakeholder groups around the perception of efficiency gains in OST in comparison to their perception of CBT. It was felt that there was reduced burden in terms of accessing OST (e.g. less clinic visits, less anxiety build-up, reduced opportunity for avoidance) and therefore a higher level of motivation to engage. These findings highlight the unique nature of treatment for this type of population and why OST may be more suited. This may be due to avoidance which is a common behavioural characteristic present in individuals with specific phobias. The format of traditionally used CBT may be unhelpful in this context due to presenting more opportunities for avoidance, inadvertently exacerbating maintaining factors and prolonging the phobia.

Whilst variations in the themes across participants groups were present, our findings from specifically focusing upon commonalities have identified some barriers, and some potential solutions, that may be helpful for the implementation of OST into services. It was evident that there were significant challenges to patient flow, at both service level and service user level. Notably, families faced many barriers in accessing support for their child's phobia, whilst in the context of services there appeared to be a lack of awareness of phobias and the existence of brief, evidence-based treatment options for this condition. There was also a lack of clarity around commissioning arrangements for phobias, significant variations in terms of where phobias fit in the care pathway, and whether or not services would treat them. As OST was found to be clinically and cost-effective [18,19] and acceptable, this demonstrates the importance of addressing these barriers so timely support can be accessed. The importance of improvements to care pathways for this population have been highlighted demonstrating an acute need to increase implementation capacity; considering local contexts, illness burden and resources. Clear guidance and communication around treatment of this condition would be welcomed to raise awareness of the impact of phobias, treatment availability and including the views of stakeholders in this process. At a service level, clarification of NICE guidelines around the treatment of specific phobias is needed to reduce confusion around where phobia treatments should be delivered. In addition to this, it is important to encourage the dissemination of the evidence generated through ASPECT to services via training. These measures will help to increase clinician understanding of the impact of specific phobias and OST, and ensure CYP are able to access support to reduce the impact and severity of their phobia earlier in its course.

In addition to identifying barriers, our findings have also highlighted important facilitating factors to enable the delivery of OST. Firstly, adopting a child-centred approach was perceived by all stakeholders as an important aspect of the success of this treatment—where clinicians gave CYP an active role and worked in collaboration, positive results were achieved. These findings highlight the importance of both the clinician's confidence and capacity to establish therapeutic rapport to create a positive treatment experience that enabled therapeutic gains in line with their goals. This was perceived as critical to the success of OST from the perspectives across all three groups. Some clinicians described their own fears or reluctance, demonstrating the importance of training and confidence building in the clinicians.

Another important facilitating factor was child readiness and suitability, particularly for CYP with comorbidities. This narrative was characterised by individuals being motivated intrinsically to work on their phobia and feeling a sense of resilience about engaging in this process. These findings highlight the complexities that may play a role in OST being an appropriate treatment option for many CYP. While clinicians acknowledge this varies on a case-by-case basis, and do not signal the need for a strict suitability criteria, there is a suggestion that development of guidance around suitability would be useful. This would support informed

decision making for CYP and clinicians around whether OST is an appropriate treatment option for them. Another facilitating aspect of success through OST was careful parental involvement; an appropriate level of parental involvement enabled CYP to maintain therapeutic gains after the session. These findings highlight the importance of successful handover to parents in order to maintain post-therapeutic gains. It was clear from the findings that this would need to be decided on a case-by-case basis, particularly in the context of parents who were advertently and inadvertently colluding with the phobia.

## Strengths & limitations

Key strengths to this research were its real-world pragmatic setting, use of purposive sampling, and an established framework to contextualise the findings in consultation with PPI. The trial was conducted across 13 sites comprising 12 NHS Trusts (including 26 CAMHS sites and three affiliated voluntary agency services) and one university-based CYP well-being service. Interviewing families from these real-world settings ensured a high level of ecological validity in our findings. Purposive sampling ensured that a range of ages, phobia types and genders were represented in our study findings. This also supported recruitment of clinicians, the majority of whom had delivered at least one OST intervention as part of the trial. This enhanced ecological validity of the findings as many clinicians had limited experience of delivering phobia interventions for CYP, therefore narratives around their confidence and skills of delivering OST may be at a similar level to those present in their broader practice community. The emergence of both positive and negative narratives across our three samples demonstrates a sensitivity to a range of participant experiences and a nuanced picture in the narratives gathered. Data coding was undertaken independently by six researchers using an established framework to contextualise the findings. These interpretations were also presented to PPI representatives to ensure relevance.

Despite these strengths, limitations to this study are apparent. We were able to successfully gather in-depth perspectives around delivery of OST but were not able to explore acceptability of the intervention received by the control group (CBT) due to resource limitations. While much of the findings focused on the format of OST, it was difficult to confirm whether those aspects of acceptability were unique to this intervention, whether these would apply to other forms of treatment, and whether people would find CBT more acceptable. Despite our recruitment pools for parent and CYP interviews being established independently at baseline, it could be argued that those who participated in an interview may have had a more positive experience of OST and this may have biased the findings. While every effort was taken to recruit participants into the study regardless of outcome, more could have been done to ensure maximum variability in our sampling in the context of engaging and non-engaging participants. A final limitation to acknowledge is vulnerability to recall bias [26] as all interviews were conducted after OST had been completed. While this maximised the validity of the main trial design, this may have meant that accounts from CYP, parents/guardians and clinicians may have been inadvertently influenced by post-hoc reconstruction of events. A flexible and creative approach was taken to support and enhance data collection with younger children, however it was apparent that narratives from older children were more detailed in content. Although there is concern this may have biased interpretations towards views of older children, synthesis across datasets has ensured an inclusive account of OST acceptability.

## Conclusion

This qualitative study examined the experiences of CYP, their parents/guardians and clinicians in receiving and delivering OST. A good level of acceptability was found across all three

samples and barriers and facilitating factors to the successful implementation of this treatment into current practice identified. Further research exploring the experiences of those who received CBT to understand how it compares to OST in terms of acceptability would be welcomed. We would also recommend further research developing suitability guidance for OST, particularly in the context of CYP with additional comorbidities, to ensure its successful implementation into services.

## Supporting information

**S1 File. Children and young people topic guide.**
(PDF)

**S2 File. Parent and guardian topic guide.**
(PDF)

**S3 File. Clinician topic guide.**
(PDF)

**S4 File. Summary coding tables.**
(PDF)

**S5 File. COREQ checklist.**
(PDF)

## Acknowledgments

The authors gratefully acknowledge the children and young people, parents/guardians and clinicians who participated in ASPECT and for sharing their experiences. We thank Joseph Horne and Alix Smith for their support in conducting and analysing interviews. We also thank our PPI representatives who informed the overall trial design, design of topic guides, and provided feedback on the findings.

## Author Contributions

**Conceptualization:** Penny Bee, Katie Biggs, Barry Wright.

**Data curation:** Emily Hayward, Kiera Solaiman, Penny Bee, Amy Barr, Hannah Edwards, Jennifer Lomas, Lucy Tindall, Alexander J. Scott, Katie Biggs.

**Formal analysis:** Emily Hayward, Kiera Solaiman, Penny Bee, Amy Barr, Hannah Edwards, Jennifer Lomas.

**Funding acquisition:** Penny Bee, Lucy Tindall, Katie Biggs.

**Investigation:** Emily Hayward, Kiera Solaiman, Penny Bee, Amy Barr, Hannah Edwards, Jennifer Lomas, Lucy Tindall, Katie Biggs.

**Methodology:** Emily Hayward, Kiera Solaiman, Penny Bee, Amy Barr, Hannah Edwards, Jennifer Lomas, Alexander J. Scott, Katie Biggs, Barry Wright.

**Project administration:** Emily Hayward, Kiera Solaiman, Penny Bee, Amy Barr, Hannah Edwards, Jennifer Lomas, Lucy Tindall, Alexander J. Scott.

**Resources:** Emily Hayward, Kiera Solaiman, Penny Bee, Amy Barr, Hannah Edwards, Jennifer Lomas, Lucy Tindall, Alexander J. Scott.

**Software:** Emily Hayward, Kiera Solaiman, Amy Barr, Hannah Edwards, Jennifer Lomas, Lucy Tindall, Alexander J. Scott.

**Supervision:** Penny Bee, Lucy Tindall, Alexander J. Scott, Katie Biggs, Barry Wright.

**Validation:** Emily Hayward, Kiera Solaiman, Penny Bee, Amy Barr, Hannah Edwards, Jennifer Lomas.

**Visualization:** Emily Hayward, Kiera Solaiman, Penny Bee, Amy Barr, Hannah Edwards, Jennifer Lomas.

**Writing – original draft:** Emily Hayward, Kiera Solaiman, Penny Bee.

**Writing – review & editing:** Emily Hayward, Kiera Solaiman, Penny Bee, Amy Barr, Hannah Edwards, Jennifer Lomas, Lucy Tindall, Alexander J. Scott, Katie Biggs, Barry Wright.

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
