## [Decision Letter · Decision Letter 0]

12 Jul 2022

PONE-D-21-37269One-Session Treatment for Specific Phobias: Barriers, facilitators and acceptability as perceived by children & young people, parents, and cliniciansPLOS ONE

Dear Dr. Hayward,

Thank you for submitting your manuscript to PLOS ONE. After careful consideration, we feel that it has merit but does not fully meet PLOS ONE’s publication criteria as it currently stands. Therefore, we invite you to submit a revised version of the manuscript that addresses the points raised during the review process.

Please respond to the minor issues raised by the reviewer. 

We look forward to receiving your revised manuscript.

Kind regards,

Xenia Gonda

Academic Editor

PLOS ONE

Journal Requirements:

3. Thank you for stating the following financial disclosure: "This research was funded by the National Institute for Health Research under its Health Technology Assessment Programme (Grant Reference Number: HTA 15/38/04). The views expressed are those of the author(s) and not necessarily those of the NIHR or the Department of Health and Social Care."

Please state what role the funders took in the study.  If the funders had no role, please state: "The funders had no role in study design, data collection and analysis, decision to publish, or preparation of the manuscript.

Reviewers' comments:

Reviewer's Responses to Questions

**Comments to the Author**

1. Is the manuscript technically sound, and do the data support the conclusions?

Reviewer #1: Yes

2. Has the statistical analysis been performed appropriately and rigorously? 

Reviewer #1: N/A

3. Have the authors made all data underlying the findings in their manuscript fully available?

Reviewer #1: No

4. Is the manuscript presented in an intelligible fashion and written in standard English?

Reviewer #1: Yes

5. Review Comments to the Author

Reviewer #1: This article reports on a qualitative sub-study as part of the ASPECT trial, to assess acceptability of one session treatment for specific phobias and common barriers and facilitators, as identified through interviews with children/young people, parents and clinicians who deliver treatment. The article clearly presents the key themes with illustrative quotations and provides sufficient detail regarding to the methodology.

The authors do not provide access to all data underlying findings. It would be helpful to provide a more detailed data availability statement regarding restrictions on availability of data in line with the data availability policy. If there is no opportunity to access data other than the quotations selected for the article and the summary coding table, the statement could usefully state this and comment on the selection of data presented in this article and supplementary material. I feel that significant data are provided in the article, and that the summary coding table is detailed and thorough, so a reader can understand how the researchers reached their conclusions. Nevertheless, I would recommend additional information in the Data Availability statement to explicitly address the issue of access to data.

6. PLOS authors have the option to publish the peer review history of their article (what does this mean?). If published, this will include your full peer review and any attached files.

Reviewer #1: No

---

## [Author Response · Author response to Decision Letter 0]

18 Aug 2022

All responses to editor and reviewer comments have been outlined on a point-by-point basis in a table in our Response to Reviewers document. Any changes requested to be outlined in our covering letter have been added. These are also documented below: 

Editor Comments:

RESPONSE: The revised manuscript and file names have been updated in line with PLOS ONE submission style templates.

We contacted the PLOS LaTex helpdesk for technical support. In our correspondence with Glenn Jackson he advised that our submission would not need to be updated as the manuscript has been built using Microsoft Word and to submit in a docx format. Therefore, we have uploaded our revised manuscript (tracked changes and clean version) in docx format. 

Thank you for stating the following financial disclosure: "This research was funded by the National Institute for Health Research under its Health Technology Assessment Programme (Grant Reference Number: HTA 15/38/04). The views expressed are those of the author(s) and not necessarily those of the NIHR or the Department of Health and Social Care."

Please state what role the funders took in the study. If the funders had no role, please state: "The funders had no role in study design, data collection and analysis, decision to publish, or preparation of the manuscript. If this statement is not correct you must amend it as needed. Please include this amended Role of Funder statement in your cover letter; we will change the online submission form on your behalf. We confirm that the statement “he funders had no role in study design, data collection and analysis, decision to publish, or preparation of the manuscript.” This has been added to our covering letter. In your Data Availability statement, you have not specified where the minimal data set underlying the results described in your manuscript can be found. PLOS defines a study's minimal data set as the underlying data used to reach the conclusions drawn in the manuscript and any additional data required to replicate the reported study findings in their entirety. All PLOS journals require that the minimal data set be made fully available. For more information about our data policy, please see http://journals.plos.org/plosone/s/data-availability.

RESPONSE: While the data presented does not contain any direct identifiers, due to the sensitive nature of the data presented and vulnerable participant group, we would not be able to make the dataset publicly available due to ethical reasons. The following wording can be added to the Data Availability statement: “Any requests for sharing of data can be made to the corresponding author or Sheffield Clinical Trials Unit (ctru@sheffield.ac.uk). The ASPECT trial management team will consider the sharing of data on a case-by case basis in line with the ethical approval and patient information sheets.” This information has been added to our covering letter.

RESPONSE: The corresponding author’s affiliation has been updated to Leeds & York Partnership NHS Foundation Trust. We have been asked by the organisation for any invoices for payment to be addressed to : FAO Sarah Cooper, Leeds & York Partnership NHS Foundation Trust, RGD Payables 4425, Phoenix House, Topcliffe Lane, Tingley, Wakefield, WF3 1WE.

RESPONSE: We have reviewed our reference list and included a list of changes in our response to reviewers letter.

5. Please amend your Response to Reviewers letter to include a point by point response to each of the points made by the Editor and / or Reviewers. Please follow this link for more information: http://blogs.PLOS.org/everyone/2011/05/10/how-to-submit-your-revised-manuscript/

RESPONSE: We have amended the letter to include a table with point by point responses to editors and the reviewer.

Thank you for stating the following financial disclosure: "This research was funded by the National Institute for Health Research under its Health Technology Assessment Programme (Grant Reference Number: HTA 15/38/04). The views expressed are those of the author(s) and not necessarily those of the NIHR or the Department of Health and Social Care."

6. Please state what role the funders took in the study. If the funders had no role, please state: "The funders had no role in study design, data collection and analysis, decision to publish, or preparation of the manuscript.

Please include this amended Role of Funder statement in your cover letter or in the Author Comments section; we will change the online submission form on your behalf.

RESPONSE: We confirm that the statement “the funders had no role in study design, data collection and analysis, decision to publish, or preparation of the manuscript.” This has been added to our covering letter. 

7. Please note that in order to use the direct billing option the corresponding author must be affiliated with the chosen institute. Please either amend your manuscript to change the affiliation or corresponding author, or email us at plosone@plos.org with a request to remove this option. Please provide your response in the "Author Comments" section.

RESPONSE: The corresponding author’s affiliation has been updated to Leeds & York Partnership NHS Foundation Trust. We have been asked by the organisation for any invoices for payment to be addressed to : FAO Sarah Cooper, Leeds & York Partnership NHS Foundation Trust, RGD Payables 4425, Phoenix House, Topcliffe Lane, Tingley, Wakefield, WF3 1WE. Our response has been provided in the Author Comments section.

8. Can you please upload an additional copy of your revised manuscript that does not contain any tracked changes or highlighting as your main article file. This will be used in the production process if your manuscript is accepted. Please amend the file type for the file showing your changes to Revised Manuscript w/tracked changes. Please follow this link for more information: http://blogs.PLOS.org/everyone/2011/05/10/how-to-submit-your-revised-manuscript/

RESPONSE: The clean version of our manuscript has been updated with all tracked changes accepted.

9. We note your current Data Availability Statement says: "While the data presented does not contain any direct identifiers, due to the sensitive nature of the data presented and vulnerable participant group, we would not be able to make the dataset publicly available due to ethical reasons. Any requests for sharing of data can be made to the corresponding author or Sheffield Clinical Trials Unit. The ASPECT trial management team will consider the sharing of data on a case-by case basis in line with the ethical approval and patient information sheets." Before we can proceed, please provide the contact information for the Sheffield Clinical Trials Unit. We will update your Data Availability Statement on your behalf once you provide the necessary information.

RESPONSE: While the data presented does not contain any direct identifiers, due to the sensitive nature of the data presented and vulnerable participant group, we would not be able to make the dataset publicly available due to ethical reasons. The following wording can be added to the Data Availability statement: “Any requests for sharing of data can be made to the corresponding author or Sheffield Clinical Trials Unit (ctru@sheffield.ac.uk). The ASPECT trial management team will consider the sharing of data on a case-by case basis in line with the ethical approval and patient information sheets.”

Reviewer Comment:

1. This article reports on a qualitative sub-study as part of the ASPECT trial, to assess acceptability of one session treatment for specific phobias and common barriers and facilitators, as identified through interviews with children/young people, parents and clinicians who deliver treatment. The article clearly presents the key themes with illustrative quotations and provides sufficient detail regarding the methodology.

RESPONSE: Thank you!

2. The authors do not provide access to all data underlying findings. It would be helpful to provide a more detailed data availability statement regarding restrictions on availability of data in line with the data availability policy. If there is no opportunity to access data other than the quotations selected for the article and the summary coding table, the statement could usefully state this and comment on the selection of data presented in this article and supplementary material. I feel that significant data are provided in the article, and that the summary coding table is detailed and thorough, so a reader can understand how the researchers reached their conclusions. Nevertheless, I would recommend additional information in the Data Availability statement to explicitly address the issue of access to data.

RESPONSE: Thank you for this feedback. We have provided updated information for the Data Availability statement to address this.

---

## [Editor Report · Decision Letter 1]

30 Aug 2022

One-Session Treatment for Specific Phobias: Barriers, facilitators and acceptability as perceived by children & young people, parents, and clinicians

PONE-D-21-37269R1

Dear Dr. Hayward,

We’re pleased to inform you that your manuscript has been judged scientifically suitable for publication and will be formally accepted for publication once it meets all outstanding technical requirements.

Kind regards,

Xenia Gonda

Academic Editor

PLOS ONE
---

## [Editor Report · Acceptance letter]

6 Sep 2022

PONE-D-21-37269R1 

One-Session Treatment for Specific Phobias: Barriers, facilitators and acceptability as perceived by children & young people, parents, and clinicians 

Dear Dr. Hayward:

I'm pleased to inform you that your manuscript has been deemed suitable for publication in PLOS ONE. Congratulations! Your manuscript is now with our production department. 

Kind regards, 

on behalf of

Dr. Xenia Gonda 

Academic Editor

PLOS ONE